# *Mycobacterium tuberculosis* PE_PGRS19 Induces Pyroptosis through a Non-Classical Caspase-11/GSDMD Pathway in Macrophages

**DOI:** 10.3390/microorganisms10122473

**Published:** 2022-12-14

**Authors:** Jianing Qian, Youwei Hu, Xiao Zhang, Mingzhe Chi, Siyue Xu, Honghai Wang, Xuelian Zhang

**Affiliations:** State Key Laboratory of Genetic Engineering, School of Life Science, Fudan University, Shanghai 200438, China

**Keywords:** PE/PPE family, host–pathogen interaction, pyroptosis, pro-inflammation

## Abstract

The PE/PPE protein family commonly exists in pathogenic species, such as *Mycobacterium tuberculosis*, suggesting a role in virulence and its maintenance. However, the exact role of most PE/PPE proteins in host–pathogen interactions remains unknown. Here, we constructed a recombinant *Mycobacterium smegmatis* expressing *M. tuberculosis* PE_PGRS19 (Ms_PE_PGRS19) and found that PE_PGRS19 overexpression resulted in accelerated bacterial growth in vitro, increased bacterial survival in macrophages, and enhanced cell damage capacity. Ms_PE_PGRS19 also induced the expression of pro-inflammatory cytokines, such as IL-6, TNF-α, IL-1β, and IL-18. Furthermore, we demonstrated that Ms_PE_PGRS19 induced cell pyroptosis by cleaving caspase-11 via a non-classical pathway rather than caspase-1 activation and further inducing the cleavage of gasdermin D, which led to the release of IL-1β and IL-18. To the best of our current knowledge, this is the first report of a PE/PPE family protein activating cell pyroptosis via a non-classical pathway, which expands the knowledge on PE/PPE protein functions, and these pathogenic factors involved in bacterial survival and spread could be potential drug targets for anti-tuberculosis therapy.

## 1. Introduction

*Mycobacterium tuberculosis*, the pathogen responsible for tuberculosis (TB), is one of the top ten causes of death worldwide. The ability of *M. tuberculosis* to manipulate the host cell fate, as host defense against bacterial clearance or as a strategy for pathogen multiplication and transmission during infection, is critical for the progression of TB. Many *M. tuberculosis* proteins are involved in host–pathogen interactions. The PE/PPE protein family is widely present in pathogenic and slow-growing mycobacteria, with *M. tuberculosis* H37Rv, including 99 PE and 69 PPE genes [1]; however, these protein-encoding genes are barely present in the non-pathogenic and fast-growing *Mycobacterium smegmatis* [2]. Therefore, the PE/PPE family is thought to be closely related to mycobacterial pathogenicity and play an important role in host–pathogen interactions.

The PE_PGRS5 protein induces cell apoptosis, which contributes to bacterial clearance and adaptive immune activation [3,4,5], whereas PE_PGRS62 [6] and PE_PGRS18 [7] inhibit host cell apoptosis. In addition, PE_PGRS41 [8] and PE_PGRS47 [9] regulate the formation of the autophagosome–lysosome to inhibit cell autophagy and enhance bacterial survival. Moreover, PPE27 induces cell necrosis and increases the bacterial survival rate under stressful conditions. A recent study showed that PPE60 is involved in the linear ubiquitin chain assembly complex-mediated nuclear factor (NF)-kB signaling pathway and regulates cell pyroptosis [10]. Increasing evidence has shown that the PE/PPE proteins play an important role in activating cellular immune responses and regulating cell fate [11]; however, the exact functions of most PE/PPE proteins have not been determined and still need to be explored.

PE_PGRS19 is encoded by the PE/PPE family gene *Rv1067* and expressed in *M. tuberculosis* H37Rv-infected guinea pig lungs at 30 and 90 d post-infection [12]. However, its role in host–pathogen interactions remains unclear. Our characterization of PE_PGRS19 using *Rv1067*-recombinant *M. smegmatis* revealed that PE_PGRS19 played an essential role in inducing cell pyroptosis and enhancing pro-inflammatory cytokine production through a non-classical caspase-11 activation pathway. This might represent a novel pathway elicited by the PE/PPE family protein of mycobacteria.

## 2. Materials and Methods

### 2.1. Construction of Recombinant M. smegmatis (Ms_PE_PGRS19)

The *Rv1067* gene encoding *M. tuberculosis* PE_PGRS19 was amplified using the primers (Sangon, Shanghai, China) listed in Table 1. The PE_PGRS19 product was digested with restriction enzymes *Bam*HI and *Hin*dIII, inserted into the kanamycin-resistant (KanR) pMV261 plasmid, and transformed into the *M. smegmatis* mc^2^155 strain. PE_PGRS19-positive bacteria were verified by Western blotting using an anti-His antibody. Subsequently, the recombinant bacteria Ms_PE_PGRS19 and the control strain Ms_Vec were cultured in 7H9 medium containing 0.2% glycerol (*v*/*v*), 0.05% Tween^®^ 80 (*v*/*v*), 50 μg/mL kanamycin, and 10% oleic acid-albumin-dextrose-catalase (OADC) supplement at 37 °C.

### 2.2. Growth and Phenotype of the Recombinant Bacteria In Vitro

The recombinant strains were incubated on 7H10-OADC plates at 37 °C for 7 d, and colony morphology, size, and shape were observed using a Zeiss Stereo-Discovery V20 microscope (Oberkochen, Germany).

The strains were grown in 7H9 liquid medium supplemented with 0.05% Tween^®^ 80 (*v*/*v*) and 0.2% glycerol (*v*/*v*) until OD600 reached 0.6–0.8; the bacteria were then collected by centrifugation at 8000 rpm for 3 min, resuspended in 50 mL fresh 7H9 liquid medium, and further cultured at 37 °C with shaking (100 rpm). The growth curve of the two strains was evaluated by measuring the absorbance at OD600 at 0 and 16 h, and then every 2–4 h until 46 h.

The recombinant strains were incubated in 7H9-OADC (KanR) media at 37 °C, harvested, washed once with phosphate-buffered saline (PBS) solution, resuspended in fixing reagent 2.5% glutaraldehyde (EM Grade, Solarbio, Beijing, China), and stained with 5% uranyl acetate and 3% lead citrate. The length and width of individual bacteria were measured via transmission electron microscopy (TEM; JEM-Z300FSC CRYO ARM™ 300, JEOL, Tokyo, Japan).

To evaluate biofilm formation, the recombinant strains were grown in 7H9-OADC (KanR) media at 37 °C and harvested when the OD600 reached 0.6. After washing twice with PBS, each strain was adjusted to an OD600 of 0.1 and diluted 1:10 in Sauton’s media. The bacteria were cultured stably at 37 °C in parafilm-wrapped 24-well plates for 4 and 10 d. Subsequently, they were observed under a Zeiss Stereo-Discovery V20 microscope.

To measure the difference in sensitivity to antibiotics between the two strains, they were cultured in 7H9-OADC(KanR) media at 37 °C until OD600 reached 0.6–0.8; then, 1 mL cells was harvested, diluted, and spread onto 7H10(KanR) plates. Afterwards, filter paper soaked with 10 μL of isoniazid (INH; 250 μg/mL and 50 μg/mL) and streptomycin (Strep; 15.625 μg/mL and 31.25 μg/mL), respectively, were placed on different regions of the plates, and the size of the inhibition area was measured after culturing them for 3 d at 37 °C.

### 2.3. Intracellular Survival Assays and Lactate Dehydrogenase (LDH) Cytotoxicity Assay

We seeded J774A.1 cells at a density of 5 × 10^5^ cells/well in 12-well culture plates and cultured them overnight in Dulbecco’s modified Eagle medium (DMEM) (Gibco, Grand Island, NY, USA) supplemented with 10% fetal bovine serum (FBS, Gibco), 100 U/mL penicillin (Gibco), and 0.1 mg/mL Strep (Gibco) at 37 °C, with 5% CO_2_. Before infection, the cells were washed twice with PBS to remove the antibiotics. The cells were infected with Ms_PE_PGRS19 and Ms_Vec at a multiplicity of infection (MOI) of 10:1 (bacteria-to-J774A.1-cell ratio) in DMEM. Four hours after infection, DMEM supplemented with 20 μg/mL gentamicin was added for 2 h to remove uninfected bacteria. At 6, 12, 24 and 48 h after the extracellular bacterial removal step, macrophages were washed three times with PBS and lysed using 0.025% sodium dodecyl sulfate (SDS). The lysed macrophages were diluted and plated on 7H10-OADC (KanR) agar plates, and colony-forming units (CFU) were determined after culturing them for 3 d at 37 °C as a measure of the intracellular survival of Ms_PE_PGRS19 and Ms_Vec.

After the clearance of extracellular bacteria, the supernatants of the J774A.1 cells infected with the recombinant strains were harvested at 6, 12, 24 and 48 h after the extracellular bacterial removal step to quantify the release of LDH from the cells. LDH release was measured using an LDH cytotoxicity assay kit (Beyotime, Shanghai, China) according to the manufacturer’s protocol. The experiments were performed in triplicate.

### 2.4. Cytokine Analysis via Enzyme-Linked Immunosorbent Assay and Quantitative Real-Time PCR

The cells were infected with Ms_PE_PGRS19 and Ms_Vec at an MOI of 10:1 in DMEM. Four hours after infection, DMEM supplemented with 20 μg/mL gentamicin was added for 2 h to remove extracellular bacteria. Then, interleukin (IL)-6, TNF-α, IL-1β and IL-18 secreted from macrophages in culture supernatants infected with Ms_PE_PGRS19 and Ms_Vec at 6, 12, 24 and 48 h were measured using enzyme-linked immunosorbent assay (ELISA) kits (IL-6, TNF-α, and IL-1β: Dakewe Biotech, Shanghai, China; IL-18: Multi Sciences, Hangzhou, China) according to the manufacturer’s instructions. After removing the supernatant, the cells were harvested and washed three times with PBS after being infected at 6, 12, 24 and 48 h. An RNAprep pure Micro Kit (Tiangen, Beijing, China) was used to extract cell RNA, and a GoScript^TM^ reverse transcription system (Promega, Madison, WI, USA) was used to obtain the cDNA after measuring the concentration and quality of RNA. The cDNA was divided and stored at −80 °C until use. In our pharmacological inhibitor experiments, the macrophages were pre-incubated for 2 h at 37 °C with the pyroptosis inhibitor necrosulfonamide (NSA) (Sigma Aldrich, St. Louis, MO, USA), and then infected with Ms_PE_PGRS19 and Ms_Vec. At 8 h after infection, the cells were harvested and RNA was extracted and reverse-transcribed. The cytokine mRNA levels were detected by real-time (RT)-PCR using the specific primers listed in Table 1 at the following cycling conditions: 95 °C for 5 min, 95 °C for 15 s and 60 °C for 30 s repeated 40 times. The experiments were performed in triplicate.

### 2.5. Cell Death Analysis by Flow Cytometry and Observation under Microscope

We seeded J774A.1 cells at a density of 1 × 10^6^ cells/well in 6-well culture plates and cultured them overnight in DMEM supplemented with 10% FBS, 100 U/mL penicillin, and 0.1 mg/mL Strep (Gibco) at 37 °C, with 5% CO_2_. After washing twice with PBS to remove the antibiotics, the macrophages were infected with Ms_PE_PGRS19 and Ms_Vec at an MOI of 20 for 4 h and those outside the cell were killed with DMEM supplemented with 20 μg/mL gentamicin for another 2 h. The cells were then cultured for 16 h and washed three times with PBS. Cell death was detected using a fluorescein isothiocyanate (FITC)-conjugated Annexin V/propidium iodide (PI) staining kit (BD Pharmingen Inc., San Diego, CA, USA) according to the manufacturer’s instructions. The cells were gently removed from the culture plates with 0.25% trypsin–ethylene diamine tetra acetic acid (EDTA) (Gibco) and carefully washed with PBS and centrifuged twice at 800× *g* to remove trypsin-EDTA. The cells were then stained with annexin V/PI and analyzed using a FACS Calibur instrument (BD Biosciences), and the data were processed using the FlowJo V10 software. All samples were analyzed within 1 h of the cells being stained by Annexin V/PI.

To observe the cell death type, J774A.1 cells were seeded in a confocal cell culture plate (Thermo Fisher Scientific, Waltham, MA, USA) and infected with Ms_PE_PGRS19 and Ms_Vec at an MOI of 10. The infected J774A.1 cells were then observed under Andor Dragonfly 200 live cell confocal.

### 2.6. Macrophage Protein Collection and Western Blot

We seeded J774A.1 cells at a density of 1 × 10^6^ cells/well in 6-well culture plates and cultured them overnight in DMEM (Gibco) supplemented with 10% FBS (Gibco), 100 U/mL penicillin (Gibco), and 0.1 mg/mL Strep (Gibco) at 37 °C, with 5% CO_2_.

Macrophages were infected with Ms_PE_PGRS19 and Ms_Vec at an MOI of 10 after washing twice with PBS, and then, they were harvested at indicated time points. The cells were treated with lipopolysaccharides (LPSs, Sigma Aldrich) at 100 ng/mL for 4 h and 10 μmol/L nigericin (Biovision) for 2 h as a positive control. The cells were lysed on ice for 20 min by adding radioimmunoprecipitation assay (RIPA) buffer (Epizyme, Shanghai, China) supplemented with a protease inhibitor cocktail (Sigma Aldrich). The lysed samples were centrifuged at 13,000 rpm and 4 °C for 10 min to remove cell fractions. Protein concentration in the supernatants was measured using a bicinchoninic acid (BCA) protein quantification assay (Aidlab Biotechnologies, Beijing, China). Equal amounts of protein were separated by 12.5% sodium dodecyl sulfate (SDS)-polyacrylamide gel electrophoresis (PAGE) and then transferred onto a 0.22-μm polyvinylidene difluoride (PVDF) membrane (Millipore, Bedford, MA, USA). The membranes were blocked with 5% non-fat milk for 1 to2 h at room temperature. After blocking, the membranes were washed three times with Tris-buffered saline with Tween^®^ 20 detergent (TBST) for 10 min each time and incubated overnight at 4 °C with primary antibodies (1:1000–1:2000), including mouse anti-His (66005-1-Ig, Proteintech, Wuhan, China), rabbit anti-caspase-11 (ab180673, Abcam, Cambridge, UK), Caspase-1 (E2Z1C) Rabbit mAb (#24232, Cell Signaling Technology, Danvers, MA, USA), Gasdermin D (E9S1X) Rabbit mAb (#39754, Cell Signaling Technology), p38 MAPK (D13E1) XP^®^ Rabbit mAb (#8690, Cell Signaling Technology), p44/42 MAPK (Erk1/2) (137F5) Rabbit mAb (#4695, Cell Signaling Technology), Phospho-p44/42 MAPK (Erk1/2) (Thr202/Tyr204) (D13.14.4E) XP^®^ Rabbit mAb (#4370, Cell Signaling Technology), Phospho-p38 MAPK (Thr180/Tyr182) (D3F9) XP^®^ Rabbit mAb (#4511, Cell Signaling Technology), Phospho-SAPK/JNK (Thr183/Tyr185) (81E11) Rabbit mAb (#4668, Cell Signaling Technology), SAPK/JNK Antibody (#9252, Cell Signaling Technology) and rabbit anti-β-actin (#8457, Cell Signaling Technology). After washing three times with Tris-buffered saline with Tween-20 (TBST) at a 1:5000 dilution ratio in 5% non-fat milk, the membranes were incubated with horseradish peroxidase (HRP)-linked goat anti-rabbit IgG secondary antibody (SA00001-2, Proteintech) for 2 h at room temperature. Specific protein bands were detected using the Super Signal West Femto Maximum Sensitivity Substrate (Thermo Scientific, Waltham, MA, USA) and a ChemiScope 3400 mini imaging system (Clinx Science Instruments, Shanghai, China). The relative intensity quantification was analyzed by imageJ v1.8.0 software and the baseline was set by blank group.

### 2.7. Statistical Analysis

Multiple *t*-tests were performed to analyze significant differences and correct for multiple comparisons using the Holm–Šídák method. The Prism 8 software (GraphPad) was used to analyze the differences between the experimental and control groups. The value *p* was considered significant at *p* < 0.05, and not significant (ns) at *p* > 0.05. * *p* < 0.05, ** *p* < 0.01, *** *p* < 0.001, and **** *p* < 0.0001. All experiments were performed in triplicate.

## 3. Results

### 3.1. PE_PGRS19 Accelerates the Growth of M. smegmatis In Vitro

To characterize PE_PGRS19, *M. smegmatis* was chosen as a fast-growing model. The full-length *Rv1067* gene with a His-tag was cloned into a pMV261 vector, and an empty pMV261 vector was transformed into the control group. Total protein lysates of the recombinant strains Ms_PE_PGRS19 and Ms_Vec were verified using Western blotting. As expected, Ms_PE_PGRS19 expressed the His-tagged PE_PGRS19 protein, whereas Ms_Vec did not (Figure 1A).

The colony size and shape of the two strains were observed using a Zeiss Stereo-microscope to characterize their phenotypes, and it was found that Ms_PE_PGRS19 had a larger size and more complicated folds than Ms_Vec after culture on 7H10 plates at 37 °C (Figure 1B). The growth rates of Ms_PE_PGRS19 and Ms_Vec were also evaluated in 7H9-OADC media, and the results showed that PE_PGRS19 overexpression accelerated the growth rate of *M. smegmetis* compared to Ms_Vec (Figure 1C). The morphology of the bacteria was further observed utilizing TEM, and it was found that bacilli length did not differ between Ms_PE_PGRS19 and Ms_Vec; however, the width increased (Figure 1D). These data suggest that PE_PGRS19 accelerated the growth of *M. smegmatis* in vitro.

As in vitro biofilms are reported to be a suitable model to investigate the persistence and drug resistance characteristics of pathogens [13], we analyzed the biofilm formation ability and antibiotic resistance of Ms_PE_PGRS19 and Ms_Vec. Compared with Ms_Vec, Ms_PE_PGRS19 exhibited faster biofilm formation in the early growth stages on Sauton’s media, but by day 10 there was no significant difference between the two strains (Figure 1E). We further compared the antibiotic resistance of Ms_PE_PGRS19 and Ms_Vec by exposure to filter paper containing different concentrations of INH (50 μg/mL and 250 μg/mL) and Strep (15.625 μg/mL and 31.25 μg/mL), respectively. The results showed that Ms_PE_PGRS19 was more resistant to INH and Strep than Ms_Vec (Figure 1F). This increase in resistance to INH and Strep was also consistent with Ms_PE_PGRS19-enhanced biofilm formation ability.

### 3.2. PE_PGRS19 Enhances the Intracellular Survival and Cell Damaging Capacity of M. smegmatis

To assess the virulence of PE_PGRS19, we measured Ms_PE_PGRS19 intracellular survival, an important prerequisite for dissemination. The invasion rate and intracellular survival ability of Ms_PE_PGRS19 significantly increased 12–24 h post-infection at a similar MOI compared with those of Ms_Vec in J774A.1 cells (Figure 2A). These two strains were further evaluated for cytotoxicity by measuring the LDH levels in the supernatants of infected macrophages. As shown in Figure 2B, PE_PGRS19 increased the ability of *M. smegmatis* to cause cell damage compared to that of the Ms_Vec-infected macrophages at the same time points in J774A.1 cells. Similar results were also observed in Raw264.7 cells infected with Ms_PE_PGRS19 and Ms_Vec (Appendix A). These data indicated that PE_PGRS19 not only enhanced bacterial invasion and intracellular survival in macrophages but also caused cytotoxicity in the infected macrophages, which are prerequisites to induce a cell immune response and regulate cell death during host–pathogen interactions.

### 3.3. PE_PGRS19 Changes the Cytokine Profile in Macrophages

We infected J774A.1 macrophages with Ms_PE_PGRS19 and Ms_Vec and analyzed the transcriptional and secretion levels of cytokines to evaluate whether the PE_PGRS19-enhanced intracellular survival and cytotoxicity were accompanied by changes in the cell cytokine profile. The results showed that PE_PGRS19 overexpression significantly enhanced the secretion (Figure 3A) and expression (Figure 3B) of pro-inflammatory cytokines such as IL-6, TNF-α, IL-1β, and IL-18. Similarly, increased expression and secretion of pro-inflammatory cytokines were also observed in Ms_PE_PGRS19-infected Raw264.7 cells (Appendix A). These results suggest that PE_PGRS19 promotes a pro-inflammatory response in the cells.

### 3.4. PE_PGRS19 Induces Cell Pyroptosis by Promoting the Cleavage of Caspase-11 and GSDMD

Ms_PE_PGRS19 infection increased LDH and pro-inflammatory cytokine release in macrophages, indicating that it could promote cell damage and cell death. To define the kind of cell death caused by Ms_PE_PGRS19 infection, J774A.1 cells were infected with Ms_PE_PGRS19 and Ms_Vec and stained with annexin V-PI; then, they were collected and subjected to flow cytometry analysis. The fluorescence-activated cell sorting (FACS) results showed an increase in PI^+^/annexin^+^ cells (Figure 4A) and a sharp decrease in PI^-^/annexin^+^ cells (Figure 4B) at 16 h post-Ms_PE_PGRS19-infection compared with Ms_Vec, suggesting that PE_PGRS19 may play a role in enhancing membrane rupture-resulted cell death and inhibiting host cell apoptosis.

Recent studies have defined a type of cell death, called pyroptosis, that is accompanied by pore formation on the plasma membrane, cell swelling, plasma membrane disruption, and release of pro-inflammatory mediators, notably IL-1 [14,15]. To determine whether PE_PGRS19-induced cell death and the release of pro-inflammatory cytokines were involved in macrophage pyroptosis, J774A.1 cells infected with Ms_PE_PGRS-19 and Ms_Vec were observed by a confocal microscope. The results showed that after 8 h of infection with Ms_PE_PGRS19, the J774A.1 cell membrane swelled and eventually ruptured, which is consistent with pyroptosis (Figure 4C) [16], suggesting that the type of cell death induced by Ms_PE_PGRS19 is pyroptosis. In addition, J774.1 cells were pretreated with the pyroptosis inhibitor NSA before infection with Ms_Vec and Ms_PE_PGRS19. Compared with Ms_Vec, Ms_PE_PGRS19 infection caused up-regulated expression of *tnf-α*, *il-6*, *il-1β*, and *il-18* genes accompanied by reduced expression of *il-10* in the absence of NSA (Figure 4D), whereas pretreatment of macrophages with NSA prior to Ms_PE_PGRS19 infection resulted in a significant decrease in the transcript levels of *il-6*, *il-1β*, and *il-18*, except for *tnf-*α (Figure 4E), which suggested that PE_PGRS19-triggered release of pro-inflammatory cytokines may be involved in macrophage pyroptosis.

Pyroptosis is characterized by the cleavage of GSDMD, which mediates the regulated lytic cell death mode and has been identified as a substrate for murine caspase-1 and caspase-11 [17]. To directly determine whether GSDMD is involved in PE_PGRS19-induced cell death, J774A.1 cells were infected with Ms_PE_PGRS19 and Ms_Vec and cell lysates were evaluated for active GSDMD cleavage products. The results showed that Ms_PE_PGRS19 induced more cleaved GSDMD N-terminal fragments (35 kDa) in macrophages after 4 and 6 h of infection compared with Ms_Vec (Figure 5A) and relative intensity data showed that the change was significant at 4 h (Figure 5B). The cleavage of caspase-11 and caspase-1 in the infected macrophages was further evaluated by Western blot using anti-caspase-11 and anti-caspase-1 antibodies, respectively. The results showed that PE_PGRS19 overexpression increased the cleavage of the p25 fragment of caspase-11 at 4 and 6 h post-infection compared with Ms_Vec (Figure 5C,D). However, caspase-1 was not activated significantly for the cleavage of caspase-1 p20 subunit by Ms_PE_PGRS19 infection (Figure 5E,F). Together, the above results demonstrate that PE_PGRS19 overexpression triggers macrophage pyroptosis and pro-inflammatory cytokine release, which was mediated by active GSDMD cleavage downstream from non-classical caspase-11 activation.

### 3.5. PE_PGRS19 Also Inhibits p38 Mitogen-Activated Protein Kinase (MAPK) Phosphorylation

MAPK pathways are also involved in the production of pro-inflammatory cytokines and the activation of macrophages [18]. Meanwhile, the activation of p38 shifts the balance of the BCL2 family towards apoptosis [19]. Thus, MAPK (p38 MAPK, JNK, and p44) phosphorylation in macrophages was further detected after infection with Ms_PE_PGRS19 and Ms_Vec via Western blot. As shown in Figure 6A,C, p38 MAPK phosphorylation was inhibited in J774A.1 macrophages after 4 h of Ms_PE_PGRS19 infection compared with Ms_Vec, whereas JNK and ERK phosphorylation had no significant and constant change indicating that PE_PGRS19 did not trigger the pro-inflammatory response in macrophages via the MAPK pathway but rather inhibited the activation of this pathway, which might have a tight relationship with the inhibition of cell apoptosis. Similar results also observed in BMDM macrophages that p38 MAPK phosphorylation was inhibited after 4 h of infection (Figure 6B,D).

## 4. Discussion

Although the PE/PPE family proteins have been in the spotlight for 20 years, the exact functions of most of them remains unclear. Given their inseparable relationship with the ESX system and their common existence in pathogenic mycobacteria, the PE/PPE proteins are believed to be involved in mycobacterial pathogenicity and regulation of the cross-talk between the host cells and pathogens.

In this study, we found that a PE_PGRS19 overexpression strain caused an increased release of the cytosolic enzyme LDH and an increased death rate of the infected cells. We further showed that the cell death type was pyroptosis as the cleaved 35 kDa GSDMD N-terminal fragment (Figure 5A) is the final and direct executor of pyroptotic cell death, and is assembled into pores, leading to the loss of ionic gradients, cell membrane rupture, and pyroptosis [20,21]. Pyroptosis is a type of proinflammatory programmed cell death mediated by caspase-1/11-cleaved GSDMD, which promotes pro-inflammatory cytokine release, notably that of IL-1β. Recent studies have reported that pyroptosis is an important component of the innate immune response to *M. tuberculosis* infections [22,23]. EST12 has been found to induce macrophage pyroptosis by binding to RACK1 and activating the NLRP3–GSDMD–IL-1β pathway through the activation of caspase-1 [24]. Our data clearly demonstrated that PE_PGRS19 caused macrophage pyroptosis leading to the release of pro-inflammatory cytokines, and the pretreatment of macrophages with the pyroptosis inhibitor NSA blocked the increased transcript levels of *il-6*, *il-1β,* and *il-18* induced by Ms_PE_PGRS19 infection. Additionally, we found that PE_PGRS19 overexpression increased the cleavage of the p25 fragment of caspase-11 compared with Ms_Vec (Figure 5C), but not the activation of caspase-1, indicating that PE_PGRS19 could trigger cell pyroptosis by activating the caspase-11–GSDMD–IL-1β pathway. Caspase-11 is an inflammatory caspase crucial to non-canonical inflammasome-mediated pyroptosis and the cleavage of GSDMD and pore-formation [21,22]. Pro-inflammatory caspase-11 triggers caspase-1-independent IL-1b and IL-18 production in response to a subset of inflammasome activators known as non-canonical activators, such as LPS of Gram-negative bacteria and cholera toxin B [25]. The caspase-11-dependent pyroptosis was also observed in mouse lung epithelial cells infected with *Burkholderia thailandensis* both in vivo and in vitro [26]. Several reports also showed that caspase-11 can protect the mice from lethal challenges with *B. thailandensis* and *Burkholderia pseudomallei* [27]. In *M. tuberculosis*, most pyroptosis-associated inflammasome activators are mediated by activity of caspase-1 [28], recently it was reported that PknF triggers the activation of both caspase-1 and caspase-11, leading to pyroptosis and pro-inflammatory cytokine release [29]. Our study indicates that PGRS19 is another novel non-canonical activator of pyroptosis in *M. tuberculosis.*

*M. tuberculosis* is a successful intracellular pathogen that evades the host’s innate immunity against chronic infections by promoting or inhibiting the host’s cellular functions [30]. In most settings, pyroptosis, autophagy, and apoptosis are thought to promote inflammation and, thereby, eliminate invading bacteria [31], while necrosis is regarded as a type of cell death for bacterial spread [32,33,34]; *M. tuberculosis* can exploit this strategy to promote its own survival. For example, *M. tuberculosis* PE_PGRS47 inhibits cell autophagy to restrict the antigen presentation of MHC-II in dendritic cells, while a mutant strain showed attenuated growth both in vivo and in vitro [9]. In addition, *M. tuberculosis Rv2626c* induced host cell necrosis, contributing to bacterial escape from the host immune surveillance [35]. However, several studies have found that some factors mediating the enhancement of cell apoptosis are used as strategies for the maintenance and dissemination of *M. tuberculosis*, such as EspC, a protein co-secreting with EsxA/B [36,37]. Additionally, PPE32 [38] and PE13 [39] functioned as promotors of host cell apoptosis, which induces bacterial intracellular survival. A recent study also demonstrated that the ESX-1-induced plasma membrane damage causes a K^+^ efflux and activation of the NLRP3-dependent IL-1β release and pyroptosis, facilitating the spread of bacteria to neighboring cells [40]. Here, we observed that PE_PGRS19 overexpression increased the bacterial load in macrophages, indicating that PE_PGRS19 may also be a virulence factor facilitating the spread of *M. tuberculosis* to neighboring cells. Indeed, caspase-1/11-mediated pyroptosis has been defined as a highly inflammatory form of programmed necrosis [41]. Thus, pyroptosis may be another double-edged sword, similar to apoptosis, critical for both the host defense and pathogen spread. In addition, PE_PGRS19 overexpression inhibited PI^−^/annexin^+^ cell populations (Figure 4B) mediating through p38 MAPK inhibition (Figure 6A,C), which may have contributed to bacterial intracellular survival in collaboration with PE_PGRS19-induced pyroptosis.

In the granuloma of patients with TB, a battleground for host–pathogen interactions with a diverse array of host cells that inhibit pathogen escape and transmission, the dynamic inflammatory response highlights the balance between protection and immunopathology as critical to the outcome of a TB infection. Proteomic analysis showed that PE_PGRS19 is present in pulmonary granulomas at both the initial stage (30 d) and 90 d post-infection [12], indicating its role in granuloma formation and maintenance. Based on our data, we propose that *M. tuberculosis* PE_PGRS19 triggers cell pyroptosis by activating the non-canonical caspase-11–GSDMD–IL-1β/18 pathway, which might also benefit the dissemination of bacteria in granulomas and contribute to the spread of bacteria to surrounding cells, representing a novel pathway elicited by the PE/PPE family protein of mycobacteria. Our results complement the current understanding of the functions of the PE/PPE proteins and provide a new perspective for the future exploration of cell death during a *M. tuberculosis* infection. However, the exact role of the PE_PGRS19 in the pathogenesis of *M. tuberculosis* and its interaction with host cells needs to be further validated by *pe_pgrs19*-deletion mutants.

## Figures and Tables

**Figure 1 microorganisms-10-02473-f001:**
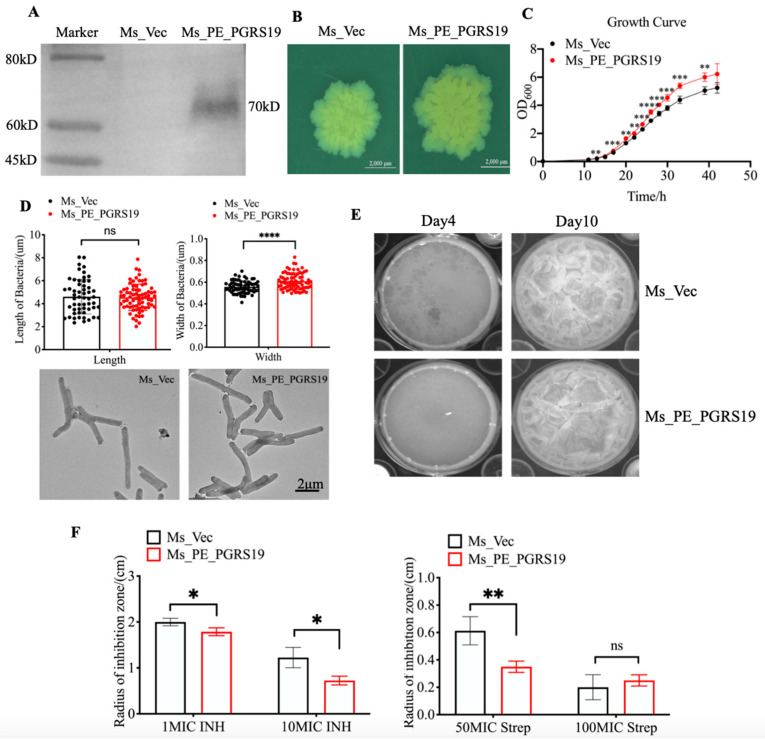
Construction and phenotype characterization of Ms_PE_PGRS19 and Ms_Vec. (**A**) Detection of the recombinant strain by Western blot using a mouse anti-His tag antibody to confirm PE_PGRS19 expression. (**B**) Colony size of Ms_PE_PGRS19 and Ms_Vec after growing on 7H10 media for 7 d. (**C**) Ms_PE_PGRS19 and Ms_Vec growth curve in 7H9-OADC media. (**D**) Bacterial length and width under TEM and *t*-test analysis. Data representative of three independent experiments with *n* > 50 cells in technical replicates per group. (**E**) Biofilm formation in Sauton’s media at 4 and 10 d. (**F**) Inhibition area of Ms_PE_PGRS19 and Ms_Vec cultured in 7H10 media with INH and Strep after 5 d. Multiple *t*-tests were performed using the Holm–Šídák method. Data presented as $\bar{x}$ ± SD of three independent experiments. ns *p* > 0.05, * *p* < 0.05, ** *p* < 0.01, *** *p* < 0.001, **** *p* < 0.0001.

**Figure 2 microorganisms-10-02473-f002:**
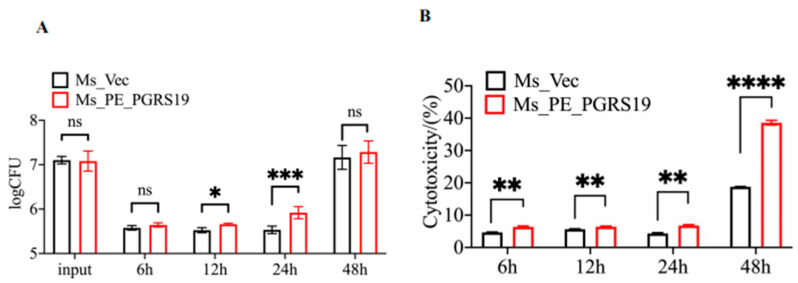
PE_PGRS19 strengthened the intracellular survival of *Mycobacterium smegmatis* and host cell toxicity. (**A**) The intracellular survival rate after infecting J774A.1 cells with Ms_PE_PGRS19 and Ms_Vec was estimated by counting CFUs at 6, 12, 24 and 48 h. (**B**) Cell toxicity was estimated by measuring LDH secretion in the supernatant of Ms_PE_PGRS19- and Ms_Vec-infected J774A.1 cells at 6, 12, 24 and 48 h. Multiple *t*-tests and 2-way ANOVA analyses were performed using the Holm–Šídák method. Data presented as $\bar{x}$ ± SD of three independent experiments. ns *p* > 0.05, * *p* < 0.05, ** *p* < 0.01, *** *p* < 0.001, **** *p* < 0.0001.

**Figure 3 microorganisms-10-02473-f003:**
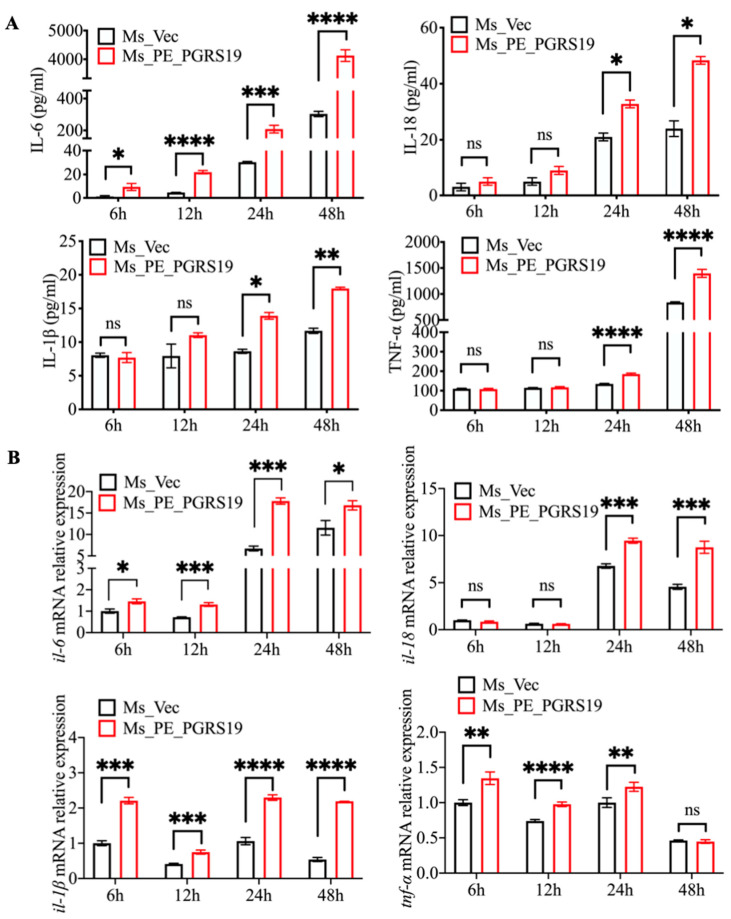
Cell cytokine expression and secretion in macrophages infected with the recombinant strains. Secretion (**A**) and mRNA relative expression (**B**) of TNF-α, IL-6, IL-1β, and IL-18 were measured by ELISA assay and real-time PCR in Ms_PE_PGRS19- and Ms_Vec-infected J774A.1 cells at 6, 12, 24 and 48 h. Data are presented as $\bar{x}$ ± SD of three independent experiments. ns *p* > 0.05, * *p* < 0.05, ** *p* < 0.01, *** *p* < 0.001, **** *p* < 0.0001.

**Figure 4 microorganisms-10-02473-f004:**
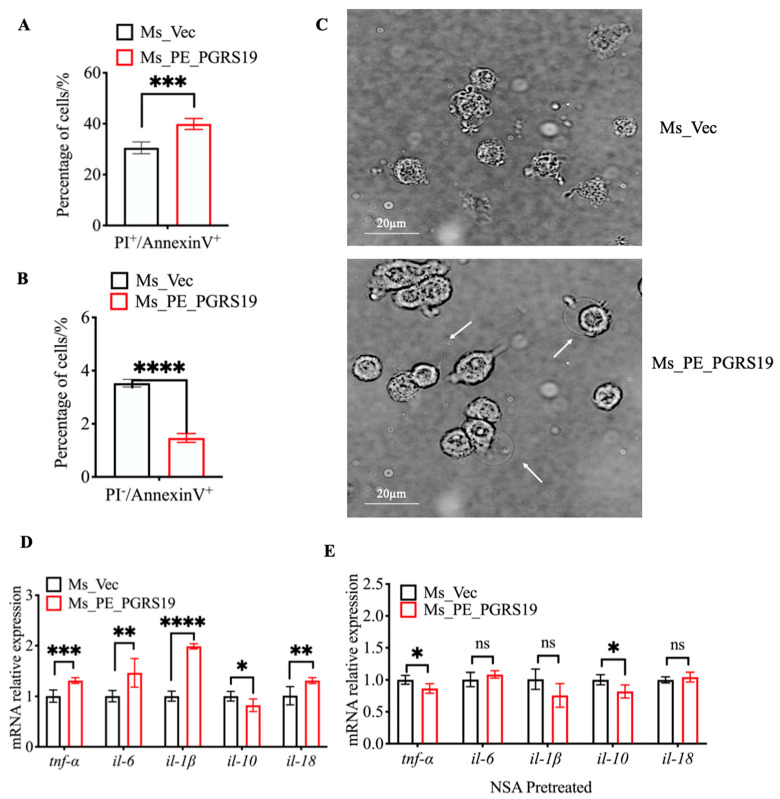
PE_PGRS19 overexpression induced macrophage membrane swelling and pro-inflammation cytokine expression, which were abolished by a pyroptosis inhibitor. J774A.1 cells were infected with Ms_PE_PGRS19 and Ms_Vec for 16 h and stained with annexin V-PI to elucidate the type of cell death. (**A**) PI^+^/annexin^+^ and (**B**) PI^−^/annexin^+^ data were analyzed using the FlowJo V10 software. (**C**) Membrane swelling was observed in J774A.1 cells under confocal microscopy after infection with Ms_PE_PGRS19 for 8 h. (**D**) mRNA relative expression levels of *il-1β*, *il-18*, *il-10*, *tnf-α* and *il-6* in J774A.1 cells at 8 h post-infection with Ms_PE_PGRS19 and Ms_Vec. (**E**) mRNA relative expression levels of *tnf-α*, *il-6*, *il-1β* and *il-18* were measured following pretreatment of macrophages with NSA before infection with Ms_PE_PGRS19 and Ms_Vec for 8 h. Data are presented as $\bar{x}$ ± SD of three independent experiments. Multiple *t*-tests were performed using with Holm–Šídák method. ns *p* > 0.05, * *p* < 0.05, ** *p* < 0.01, *** *p* < 0.001, **** *p* < 0.0001.

**Figure 5 microorganisms-10-02473-f005:**
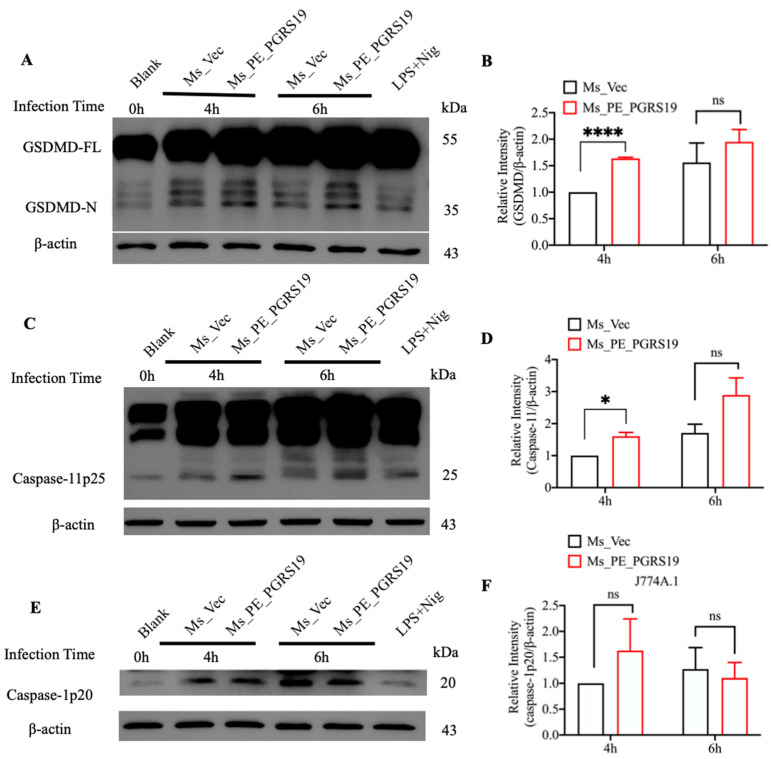
PE_PGRS19 overexpression upregulates the cleavage of GSDMD and caspase-11 in J774A.1 cells. Whole-cell lysate proteins were detected to measure the cleavage of (**A**) GSDMD and (**B**) relative intensity at 0, 4 and 6 h post-infection with Ms_PE_PGRS19 and Ms_Vec in J774A.1 cells. The cleavage of (**C**) caspase-11 and (**D**) relative intensity was measured, as well as (**E**) caspase-1 and (**F**) relative intensity. LPS treatment (4 h) and nigericin stimulation (2 h) were used as positive controls to induce pyroptosis. Multiple *t*-tests were performed using with the Holm–Šídák method. ns *p* > 0.05, * *p* < 0.05, **** *p* < 0.0001. Data are presented as $\bar{x}$ ± SEM of three independent experiments.

**Figure 6 microorganisms-10-02473-f006:**
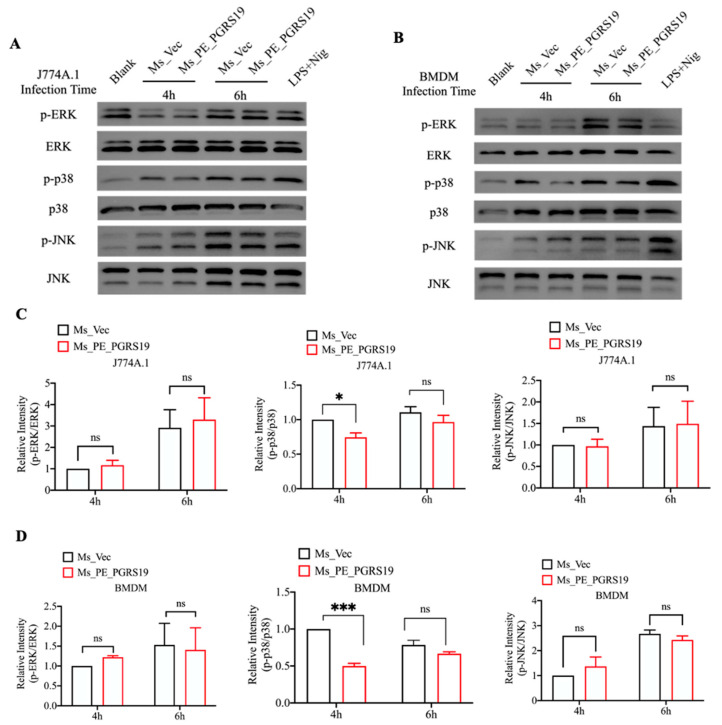
Inhibition of the p38 pathways in Ms_PE_PGRS19-infected macrophages. After infection with Ms_PE_PGRS19 and Ms_Vec at 0, 4 and 6 h, the phosphorylation levels (**A**) and relative intensity (**C**) of MAPK pathway proteins ERK, p38 and JNK were detected in J774A.1. The phosphorylation levels (**B**) and relative intensity (**D**) of MAPK pathway proteins ERK, p38 and JNK were detected in BMDM cells. LPS treatment (4 h) and nigericin stimulation (2 h) were used as positive controls. The relative intensity of blank group was set as a baseline for quantification and analyzed by image J. Data are presented as $\bar{x}$ ± SEM of three independent experiments. * *p* < 0.05, *** *p* < 0.001.

**Table 1 microorganisms-10-02473-t001:** Primers sequences used in this article.

Primers	5′-3′ Sequence	3′-5′ Sequence
Rv1067	GCAATGGCCAAGACAATTGCGTGTCGTTTGTGTTGGTGTC	TTAACTACGTCGACATCGATTTAGTGGTGGTGGTGGTGGTGCTGCCCCGGCGTGCCGGCGT
β-actin	ATTACTGCTCTGGCTCCTA	CAAGACAAGATGGTGAATGG
il-6	ATCATACTCTCCAGATACATCC	GTTCATAGCAGCCTTATTCATA
il-12	GAATGGCGTCTCTGTCTG	GCTGGTGCTGTAGTTCTC
il-1β	TCGTGAATGAGCAGACAG	ATCAGAGGCAAGGAGGAA
il-18	CCTGCCTTCTTCCTCATTCTTG	AACCTGCTGTCTGCTTCTGT
il-10	CTGCTAACCGACTCCTTAATGC	CTTGACTGCTGGCGATATGC
tnf-α	AGTGACAAGCCTGTAGCCC	GAGGTTGACTTTCTCCTGGTAT

## Data Availability

Not applicable.

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
