# Peer review of "Mycobacterium tuberculosis PE_PGRS19 Induces Pyroptosis through a Non-Classical Caspase-11/GSDMD Pathway in Macrophages"

_microorganisms, 2022, doi:10.3390/microorganisms10122473_

Round 1

Reviewer 1 Report (Previous Reviewer 2)

The authors made adequate revisions to the original comments/suggestions. 

Reviewer 2 Report (New Reviewer)

Title: Mycobacterium tuberculosis PE_PGRS19 induces pyroptosis 2 through a non-classical caspase-11/GSDMD pathway in macro-3 phages

This study by Qian J, Hu Y et al., described macrophage death by PE_PGRS19 caused by non-classical caspase-11/GSDMD-dependent pyroptosis. 

The manuscript is nicely written and the results are well described.

I have only one concern; Figure 4-C image background contains many identical spots. Could authors explain this? if necessary I would suggest providing better representative images. 

This manuscript is a resubmission of an earlier submission. The following is a list of the peer review reports and author responses from that submission.

Round 1

Reviewer 1 Report

The paper “Mycobacterium tuberculosis PE_PGRS19 enhances cell pyroptosis and the pro-inflammatory response by inducing glycolysis in macrophages” by Qian et al addresses an important question of the function of M. tuberculosis PE_PGRS19 protein and its interaction with the host. The authors demonstrated that M.tuberculosis bearing recombinant PE_PGRS19 show enhanced growth characteristics, induced pro-inflammatory cytokines production and modified macrophage metabolism towards glycolysis. 

The paper might be of interest, however there are several serious shortcomings. 

1.     My major concern is the setup of cell culture experiments. The authors demonstrated the difference between Ms_Vec and Ms_PE_PGRS19 in regard of Streptomycin sensitivity. On the figure 1 it is shown that Ms_PE_PGRS19 is more resistant to Streptomycin. At the same time the cell culture experiments were conducted in the medium containing Strep at the concentration of 0.1 mg/ml. This difference in Strep sensitivity could have biased the results of cell culture experiments. The authors have to repeat the cell culture experiments in Strep-free conditions.

2.     It is unclear why the authors used different cell lines for different experiments. Infection efficiency and bacterial survival was analyzed in RAW264.7 cells, while signaling was analysed in J774 cells. I recommend to repeat all experiments done on RAW264.7 on J774 cells for the sake of consistency. The data obtained on RAW cell can be removed then.

3.     The results, presented on figure 4 C and D are not really convincing. The authors have to provide densitometry data and properly normalize the bands, otherwise the conclusions overstate the results. As well the results obtained with BMDM do not correspond to that obtained with J774 cells.

Minor points.

4.     Figures need some editing. For example, on Figure 5A left panel and right panel differ in the sequence of the legend (on the left side Vec is in the bottom line, on the right side the Vec is in the top line), what confuses the reader. 

5.     Individual data points on bar diagrams are not needed and can be removed. 

Reviewer 2 Report

 The manuscript of Qian et al. investigates the effects of infecting murine macrophage-like cell lines (RAW264.7 and J774) with a Mycobacterium smegmatis strain expressing M. tuberculosis PE_PGRS19 protein on cell viability, inflammatory responses, and glycolysis-associated gene expression. The authors find that, compared with cell infected with a vector-transfected Mycobacterium smegmatis, infection with PE_PGRS19-overexpressing strain causes enhanced loss of cell viability and elevated pro-inflammatory gene expression and cytokine secretion. Based on caspase-1, -11, and gasdermin D cleavage, the authors suggest that macrophages die by pyroptosis. Furthermore, authors observe that some glycolytic genes are increasingly expressed in cells infected with PE_PGRS19-overexpressing strain whereas treating cells with glycolysis inhibitor 2-deoxyglucose prevents inflammatory response and cell death.

In my opinion, the major claim of the manuscript is not supported by the data presented.

Most importantly, modest changes of glycolysis-associated mRNAs, which are not consistently observed at different time points post infection, do not show that glycolysis itself is activated. This could be measured by extracellular flux analyses or directly by measuring lactate release or glucose consumption. 2-deoxyglucose at 20mM may hamper macrophage metabolism in general (see Wang et al., PMID 30184486) and cannot be used to show that enhanced glycolysis is the reason for the pyroptosis induction.

Second, up-regulation of cytokine mRNA expression after infection with PE_PGRS19-overexpressing strain is very modest at best (or not observed at all if comparing with 0h time point), so the claim of cytokine storm after the infection is very exaggerated.

These, and other specific issues (pointed below) prevent publication of the manuscript in this form.

Specific comments:

Figure 1C – growth acceleration is difficult to claim since the differences become observable at high OD values – no differences below OD of 1 were observed.

For all western blots, quantification of multiple independent experiments should be provided.

Fig 5e, the effect of zVAD should be shown for vector- and PE_PGRS19-transfected cells.

6e Effects of 2-dg alone are not shown.

Minor:

Line 303 change MARK to MAPK.

Line 328 and thereafter references to Fig4 should be to Fig5.